# Involvement of the Protein Ras Homolog Enriched in the Striatum, Rhes, in Dopaminergic Neurons’ Degeneration: Link to Parkinson’s Disease

**DOI:** 10.3390/ijms22105326

**Published:** 2021-05-18

**Authors:** Marcello Serra, Annalisa Pinna, Giulia Costa, Alessandro Usiello, Massimo Pasqualetti, Luigi Avallone, Micaela Morelli, Francesco Napolitano

**Affiliations:** 1Department of Biomedical Sciences, Section of Neuroscience, University of Cagliari, 09042 Cagliari, Italy; marcelloserra@unica.it (M.S.); giulia.costa@unica.it (G.C.); morelli@unica.it (M.M.); 2National Research Council of Italy (CNR), Neuroscience Institute—Cagliari, Cittadella Universitaria, 09042 Cagliari, Italy; apinna@unica.it; 3Department of Environmental, Biological and Pharmaceutical Sciences and Technologies, University of Campania Luigi Vanvitelli, 81100 Caserta, Italy; usiello@ceinge.unina.it; 4Laboratory of Behavioral Neuroscience, Ceinge Biotecnologie Avanzate, 80145 Naples, Italy; 5Unit of Cell and Developmental Biology, Department of Biology, University of Pisa, 56127 Pisa, Italy; massimo.pasqualetti@unipi.it; 6Department of Veterinary Medicine and Animal Productions, University of Naples “Federico II”, 80137 Naples, Italy; avallone@unina.it

**Keywords:** substantia nigra, mTOR, SUMO E3 ligase, Huntington’s disease, 3,4-methylenedioxymethamphetamine (MDMA), autophagy, L-Dopa-induced dyskinesia (LID), mitophagy

## Abstract

*Rhes* is one of the most interesting genes regulated by thyroid hormones that, through the inhibition of the striatal cAMP/PKA pathway, acts as a modulator of dopamine neurotransmission. *Rhes mRNA* is expressed at high levels in the dorsal striatum, with a medial-to-lateral expression gradient reflecting that of both dopamine D_2_ and adenosine A_2A_ receptors. *Rhes* transcript is also present in the hippocampus, cerebral cortex, olfactory tubercle and bulb, substantia nigra pars compacta (SNc) and ventral tegmental area of the rodent brain. In line with *Rhes*-dependent regulation of dopaminergic transmission, data showed that lack of *Rhes* enhanced cocaine- and amphetamine-induced motor stimulation in mice. Previous studies showed that pharmacological depletion of dopamine significantly reduces *Rhes* mRNA levels in rodents, non-human primates and Parkinson’s disease (PD) patients, suggesting a link between dopaminergic innervation and physiological *Rhes* mRNA expression. Rhes protein binds to and activates striatal mTORC1, and modulates L-DOPA-induced dyskinesia in PD rodent models. Finally, Rhes is involved in the survival of mouse midbrain dopaminergic neurons of SNc, thus pointing towards a Rhes-dependent modulation of autophagy and mitophagy processes, and encouraging further investigations about mechanisms underlying dysfunctions of the nigrostriatal system.

## 1. Discovery of Rhes

### 1.1. Protein Structure

The Ras homolog enriched in striatum (Rhes) is a 266 amino-acid (aa) protein, discovered by a subtractive hybridization procedure, in the attempt to identify striatal-enriched transcripts [1]. As the name implies, Rhes belongs to the superfamily of Ras proteins and, as such, it is made up of five G box domains, all of them normally required for the interaction with phosphate moieties of guanosine triphosphate/diphosphate (GTP/GDP) Ras-GTPase activating protein effector, and guanine nucleotide moiety [2]. Together with Dexras1, Rhes differs from other cognate members for having peculiar N- and C-terminal domains [3,4]. In this respect, while the N-terminal sequence, encompassing 1–18 amino acids, is likely to have the binding motif for the deubiquitinating enzyme, the C-terminal cationic domain interacts with Gβ_1_, Gβ_2_ and Gβ_3_ subunits of heterotrimeric G proteins [5], and contains a well-conserved CAAX motif that, following the enzymatic post-translational modification (farnesylation), is able to translocate this small protein to the plasma membrane [6,7,8].

### 1.2. Anatomical Brain Localization

*Rhes* mRNA was detected in virtually all GABAergic medium spiny projection neurons (MSNs) of rodent and human brains, as well as in mouse large aspiny cholinergic interneurons (ChIs), but not in GABAergic parvalbumin- and neuropeptide Y-positive interneurons of the mouse striatum [9,10,11]. The expression of *Rhes* was reported to be higher in the dorsal striatum than the ventral striatum (nucleus accumbens), with a peculiar medial-to-lateral gradient of increasing expression observed in both young (from 6-day-old) and adult rodents [10,12,13], thus mirroring the striatal expression pattern of both dopamine D_2_ receptor (D_2_R) and adenosine A_2A_ receptor (A_2A_R) as well [10,14]. In addition to the initial studies about its striatal localization, *Rhes* mRNA was also detected in several other areas of the central nervous system, such as the cornu Ammonis (CA) of the hippocampus (i.e., CA1, CA2 and CA3 subfields), cerebral cortex (layers II and III), piriform cortex, olfactory tubercle, subiculum, thalamus, inferior colliculus, substantia nigra pars compacta (SNc) and ventral tegmental area (VTA) of the rodent brain [4,13,15]. Similarly, in the human brain, *Rhes* transcript was observed in the hippocampal dentate gyrus and in the pyramidal cell layer of CA1, CA2 and CA3 fields [10], as well as in frontal cortical areas (layers II–VI), with the highest expression observed in layer V of the cerebral cortex [16]. More detailed studies, somehow supporting and extending such findings, were recently performed by Ehrenberg and colleagues, who documented that, using multiplex immunofluorescence and single nucleus RNA-sequencing approaches in human brain, *Rhes* is widespread in cortical neurons, CA1 pyramidal neurons, superior frontal gyrus and entorhinal cortex, where it presents an almost total diffuse cytoplasmic distribution [6]. Nonetheless, additional studies are required to assess *Rhes* expression in the human midbrain at the level of the SNc.

## 2. Ontogeny of Rhes and Its Striatal Regulation

### 2.1. Rhes Is Modulated by Thyroid Hormones

The first gene expression study aimed to evaluate the ontogeny of *Rhes* mRNA in rats was carried out by Falk and colleagues in the 1999, who documented low levels of between embryonic day 16 (E16) and postnatal day 10 (P10), while a seven-fold increase occurred between P10 and P15 [7], and stabilized from that time on [13]. This peculiar *Rhes* expression pattern mirrors that of thyroid hormones’ occurrence and prompted researchers to investigate about the putative functional correlation between *Rhes* and thyroid hormones. In this respect, Northern blot and in situ hybridization analyses, carried out in the striatal samples of congenital hypothyroid rats, revealed levels of *Rhes* mRNA as barely detectable, which were normalized following the physiological thyroxine (T4) supplementation, either by a single or repeated 3,3′,5-triiodo-L-thyronine (T3) injections [7,17,18,19]. Interestingly, no *Rhes* transcript changes were observed in adult onset of hypothyroidism in rats [19], whereas adult hypothyroid mice showed a significant reduction in striatal *Rhes* transcript [18]. Again, administration of the selective thyroid hormone receptor-beta (TRβ) agonist GC-1 was able to normalize striatal *Rhes* mRNA in congenitally hypothyroid 17-day-old rats, suggesting a significant contribution of TRβ in *Rhes* expression [17]. However, a later study in mice highlighted a major role for thyroid hormone receptor-alpha (TRα), as T3 supplementation was able to rescue striatal *Rhes* transcript exclusively in TRβ-deficient animals, but not in TRα-deficient ones [18].

### 2.2. Rhes Expression Is Regulated by Dopamine Innervation

Besides thyroid hormones, other evidence outlined a role played by dopamine innervation in regulating striatal *Rhes* mRNA in adult rodents. Accordingly, dopamine depletion, induced either by the dopaminergic/noradrenergic neurotoxin 6-hydroxydopamine (6-OHDA) or reserpine, significantly reduced *Rhes* mRNA levels throughout the striatum and olfactory tubercle of adult rats [12], while no main effect was observed in 6-OHDA-lesioned neonatal animals [13]. Consistent with observations drawn from rodents, Napolitano and coworkers also reported a significant reduction of *Rhes* mRNA levels in the striatum of both 1-methyl-4-phenyl-1,2,3,6-tetrahydropyridine (MPTP)-intoxicated non-human primates (Macaca mulatta) and Parkinson’s disease (PD) patients [20]. Overall, these findings suggest a link between intact dopaminergic innervation and physiological *Rhes* mRNA expression and, in turn, unveil a potential involvement of this small GTPase in PD pathophysiology.

## 3. Rhes Intracellular Signaling

### 3.1. In Vitro Modulation of Rhes-Dependent cAMP/PKA Signaling

The first insight about biochemical properties of Rhes was provided by Vargiu and collaborators (2004), who documented that in undifferentiated PC12 cells, Rhes seems to be active even under resting conditions, although with a low intrinsic GTPase activity, since more than 30% of this protein resulted bound to GTP [4]. Moreover, the same authors found that co-transfection of Rhes, either with thyrotropin-stimulating hormone receptor (TSHR), or with constitutively activated β_2_-adrenergic receptors, significantly inhibited the cyclic adenosine monophosphate (cAMP)/phosphate kinase A (PKA) activity. Of interest, Rhes did not directly interfere with the function of either G_αs/olf_ protein or PKA, suggesting an upstream site of action, most likely between GPCR localization and heterotrimeric G protein complex. In agreement with this view, it was later reported the ability of Rhes to affect in vitro the drug-stimulated activation of the dopamine type 1 receptor (D_1_R), with a significant reduction of cAMP accumulation and the downstream-related signaling [21]. Alongside its ability to negatively modulate GPCR signaling, further experiments in HEK293 and COS-7 cells showed that Rhes reduces Gαi-dependent signaling, by inhibiting tonic voltage-dependent Ca_V_2.2 (N-type) calcium channels, in a pertussis toxin (PTX)-dependent manner [21,22].

### 3.2. Rhes Affects Striatal cAMP/PKA Signaling in Mice

Consistent with observation performed in vitro, a negative modulatory role of Rhes over striatal D_1_R-dependent cAMP/PKA signaling in mice was reported. In this respect, administration of SKF 81297, a selective dopamine D_1_R agonist, caused a greater increased phosphorylation state of the PKA-dependent activation site Ser-845 residue of the glutamate α-amino-3-hydroxy-5-methyl-4-isoxazolepropionic acid (AMPA) receptor subunit, in *Rhes* knockout (KO) mice, when compared to wild-type (WT) controls [9,10]. Besides D_1_R, Rhes can also counteract striatal D_2_R-mediated signaling, as demonstrated by the reduced ability for dopamine to activate G_i/o_ protein, in striatal slices from *Rhes* KO mice [9]. In line with the signaling properties of Rhes in regulating dopaminergic transmission, more recent investigations showed that lack of *Rhes* significantly enhanced amphetamine-induced motor stimulation in KO mice, most likely also through the inhibitory control of the striatal-enriched guanine nucleotide exchange factor (GEF), RasGRP1, over Rhes activity [16,23]. In keeping with this, Napolitano and colleagues showed that Rhes profoundly impacted on molecular and motor stimulant effects mediated by cocaine administration. Indeed, mice lacking the *Rhes* gene showed an abnormally higher motor response to this psychostimulant in *Rhes* KO mice than WT-treated animals. Moreover, remarkable changes in cocaine-dependent protein expression were reported in KO animals within whole striatal proteome, when compared to controls [24]. Altogether, these results suggest that Rhes might act as a physiological molecular brake for the striatal dopamine responses, under phasic conditions [20,25].

### 3.3. Rhes Affects the PI3K/Akt Signaling Pathway

Early experiments, carried out in HeLa cells, indicated that Rhes functionally binds to the catalytic p110 subunit of PI3K, and when co-transfected in Cos-7 cells with Akt, it promotes Akt-mediated phosphorylation of histone H2B [4]. These findings were later confirmed and extended in HEK293T and PC12 cells where, following the treatment with different growth factors (IGF-1, EGF or PDGF), Rhes enhanced p85-PI3K interaction and, interestingly, targeted Akt to the plasma membrane, thus arguing that Rhes may function as a critical bridge between PI3K and the AKT pathway [26]. In line with in vitro data, lack of *Rhes* results in profound alteration in the excitability of ChIs, where the stimulation of D_2_R triggered an aberrant increase of action potential discharge, which was prevented by the pre-incubation with either the selective Ca_V_2.2 Ca^2+^ channel blocker, ω-conotoxin, or PI3K inhibitor, LY294002, pointing towards a functional modulation of Rhes on the PI3K/Akt signaling pathway in these neurons [11]. On the other hand, in vivo studies performed in *Rhes* KO mice demonstrated that lack of *Rhes* increased phosphorylation of Akt and glycogen synthase kinase 3 beta (GSK3-β), upon apomorphine treatment, assuming that this small molecule may be necessary to promote Akt dephosphorylation [27]. Moreover, the same authors documented that Rhes interacts with β-arrestin [27], a scaffolding protein, which is established to modulate the D_2_R-dependent Akt/GSK3-β signaling [28].

## 4. Rhes Involvement in Huntington Disease and L-DOPA-Induced Dyskinesia

### 4.1. Rhes Acts as SUMO E3 Ligase for the Mutant Huntingtin

Small ubiquitin-like modifier proteins (SUMO) represent a category of molecules covalently attached to specific lysine target residues, thus allowing changes in their localization, stability and activity, by means of a dynamic process, known as SUMOylation [29]. Interestingly, given its relevant impact on the modulation of synaptic plasticity, SUMOylation has also been implicated in a variety of neurological disorders, including PD, Huntington’s disease (HD) and amyotrophic lateral sclerosis (for a review, refer to Anderson et al., 2017 [30]). In this respect, compelling evidence pointed out that Rhes acts as SUMO E3 ligase in the striatum and, by doing so, it may participate in the HD pathogenesis, as well as in tau pathology [8,31,32]. Specifically, it was demonstrated the ability of Rhes to less avidly bind to WT huntingtin (wtHtt), and drastically increase the disperse (cytotoxic) form of mutant huntingtin (mHtt), as compared to the aggregated (cytoprotective) one, in different cellular settings [32,33]. Additionally, Rhes participates in the SUMOylation process throughout the striatum, by promoting the “cross-SUMOylation” of E1 and Ubc9 (E2) proteins, thus influencing several signaling pathways [34].

### 4.2. Role of Rhes in Modulating HD-Dependent Phenotypes in Animal Models

In agreement with the above-mentioned in vitro findings, in vivo evidence strengthened the potential involvement of Rhes in HD, since lack of *Rhes* prevented the striatal injury and motor dysfunctions in *Rhes* KO mice, induced by the mitochondrial complex II inhibitor, 3-nitropropionic acid (3-NP) [35]. Moreover, *Rhes* gene deletion either delayed or ameliorated behavioral and anatomical HD-related phenotypes in the transgenic mouse models of HD, R6/1 and B6.129P2-^Htttm2^Detl/150J, which display about 115 CAG repeats of the human mHtt allele and just the N-terminal fragment of mHtt, respectively [36,37]. Interestingly, investigations performed in R6/2 and 140 CAG knock-in HD mouse models revealed that the Golgi protein acyl-CoA binding domain containing 3 (ACBD3) and the huntingtin-associated protein 1 (Hap1) oppositely modulated Rhes E3 ligase activity, either increasing or reducing Rhes-mediated SUMOylation of mHtt [38,39]. Rhes has been recently regarded as an inducer of tunneling nanotubes (TNT)-like protrusions, which allow the communication of neighboring cells, as well as transport of the selective membrane vesicles and organelles, including mHtt rather than wtHtt [40]. Accordingly, studies of differential interference contrast microscopy, carried out in the striatal STHdhQ7/Q7 cells, demonstrated that, out of 70% of GFP-Rhes-positive cells showing filopodia-like protrusions, 30% of them exhibited TNT-like structures, thus highlighting a novel ability for Rhes to modulate striatal HD vulnerability [40]. It is worth underlying that the SUMO E3 ligase activity domain of Rhes (171–266 aa) promotes the biogenesis of TNT-like tunnels, even if only the full-length *Rhes* WT protein can be transported from cell to cell [40].

### 4.3. Rhes Affects L-DOPA-Induced Dyskinesia (LID) in PD Mouse Model

Furthermore, both in vitro and in vivo experiments showed that Rhes physiologically binds to and activates the mTOR complex 1 and 2 (mTORC1 and mTORC2, respectively), in a GTP-dependent manner [23,31]. Among a variety of cellular and molecular processes, mTORC1 has been regarded as one of the master regulators of L-3,4-dihydroxyphenylalanine (L-DOPA)-induced dyskinesia (LID) in PD rodent models [41,42]. The most efficacious symptomatic treatment in PD is the dopamine replacement with the dopamine precursor, L-DOPA. However, long-term L-DOPA therapy is associated with the development of motor complications, such as LID, which severely compromise the beneficial effects of the drug, thus becoming treatment-limiting [43,44,45,46,47]. Among different molecular changes underlying LID onset and severity [44,48], mTORC1 activation within D_1_-expressing striatal neurons following chronic L-DOPA treatment has been regarded as a key player in the modulation of such motor disturbances [42]. Accordingly, mTORC1 inhibition, either by rapamycin or rapamycin ester CCI-779, significantly reduced LID in 6-OHDA-lesioned PD rodent models, without affecting the anti-akinetic effect of L-DOPA [41,42]. In this view, studies in striatal cell lines, striatal tissue and HEK293 cells as well, documented that Rhes has the ability to selectively bind to and activate mTOR [49]. Remarkably, lack of *Rhes* significantly reduced LID occurrence and severity in 6-OHDA-lesioned KO mice, and prevented the rise of nigral GABA and glutamate release in the substantia nigra pars reticulata (SNr), which represents the output nucleus of the basal ganglia [49,50]. More recently, a direct influence of Rhes on RasGRP1-dependent signaling in promoting LID expression has been reported in animal models [51]. Overall, considering, on one hand, the potential toxicity of rapamycin and related drugs as inhibitors of protein synthesis and, on the other, the negligible levels of Rhes in peripheral tissues, these findings pave the way toward a potential use of Rhes inhibitors in PD therapy to counteract LID, with no detrimental impact on L-DOPA efficacy (Figure 1). However, in order to potentially translate this therapeutic strategy to PD patients, further new studies in *Rhes* conditional knockout mice, aimed at selectively deleting the *Rhes* gene in the nigrostriatal pathway, are mandatory, so as to better disclose the role of this small GTP-binding protein in such severe motor disturbances.

## 5. Involvement of Rhes in Parkinson’s Disease: Focus on Rhes Regulation of Nigrostriatal Neurons’ Survival

### 5.1. Rhes Counteracts Nigrostriatal Degeneration during Ageing in a Gender-Dependent Manner

The pathophysiology of PD relies on the degeneration of dopaminergic neurons located in the SNc (which project to the motor part of the striatum, caudate-putamen nucleus in humans), as well as cytoplasmic accumulation of α-synuclein-containing Lewy bodies [43,52]. Based on the occurrence of *Rhes* transcript in the midbrain tyrosine hydroxylase (TH)-positive neurons of SNc and VTA (Figure 2) [15] and, considering its role in regulating survival-related AKT and mTOR signaling pathways, further studies sought to investigate whether Rhes could also have an impact on midbrain dopaminergic neurons’ survival, under both physiological and pathological conditions. Interestingly, lack of *Rhes* led to a mild, although significant, reduction of midbrain TH-positive neurons in both 6- and 12-month-old KO male mice [15]. As a behavioral correlate to what was observed at the morphological level, mutant male animals showed significant alterations at the beam-walking test, in an age-dependent manner, taking longer to traverse the beam, thus suggesting that Rhes might drive the nigrostriatal pathway toward a susceptibility to cell death, triggered either by aging processes or by environmental toxins [15]. The mechanisms responsible for the PD-related neuronal degeneration are still elusive, and often controversial. However, several factors, such as gender, neuroinflammation, oxidative stress, excitotoxicity, reduced expression of trophic factors, and dysfunction of the protein degradation system, may influence the nigrostriatal pathway degeneration [53,54]. Although several causative genes of either dominant or recessive inherited PD forms have been identified, most of them are not yet characterized, hence requiring further studies aimed at clarifying the interplay between genetics and other possible pathogenic factors [55]. Yet, epidemiological investigations indicated that among the factors that may influence the neuronal dopaminergic degeneration, gender may have a prominent role, since males are at higher risk than females, who might be more protected by estrogens, in particular 17β-estradiol [56,57]. Therefore, based on the involvement of the α-synuclein-mediated microglia activation in PD pathogenesis [58,59,60], and considering the influence of Rhes upon the survival of nigrostriatal dopaminergic neurons [15], in a recent study by Costa and colleagues, the potential role of this protein on the inflammatory response was initially investigated during the physiological brain aging, in both male and female *Rhes* KO mice [61]. Immunohistochemistry evaluations confirm the decrease in TH immunoreactivity already observed in the midbrain nigrostriatal neurons of male *Rhes* KO mice [15], but a decrease in TH immunoreactivity was also observed in female *Rhes* KO mice. Interestingly, a higher number of the complement type 3 receptor (CD11b), as well as glial fibrillary acid protein (GFAP), were found in male rather than female KO mice [61].

### 5.2. Rhes Reduces the MDMA-Induced Dopaminergic Degeneration and Neuroinflammation Affecting the Nigrostriatal System

Among the amphetamine-related drugs, 3,4-methylenedioxymethamphetamine (MDMA, also known as ‘ecstasy’) is one of the most heavily abused psychostimulants among adolescents and young adults [62,63,64]. MDMA has addictive properties and may elicit neurotoxic effects and glia activation in several animal species, although the impact on the neural system may differ depending on the considered species [65,66,67,68]. In particular, administration of MDMA to mice triggers a peculiar profile of neurotoxicity and glia activation that involves the nigrostriatal and mesolimbic dopamine systems [63,69,70,71,72]. These interesting results allowed us to evaluate whether *Rhes* KO mice showed higher vulnerability to MDMA-dependent neurotoxic and neuroinflammatory effects in the nigrostriatal system as compared to WT animals, and whether gender and/or age might be associated with these effects. In line with this, one of the studies recently performed by Usiello and co-workers demonstrated that acute-repeated MDMA administration in adult (3-month-old) and middle-aged (12-month-old) male and female *Rhes* KO mice caused a significant dopaminergic neurodegeneration and glia activation, which was generally more pronounced in males than females [73]. In adult males, MDMA administration induced in both WT and KO animals a decrease of TH-positive fiber density in the dorsal striatum, as well as of the total number of TH-positive neurons in SNc. Conversely, female *Rhes* WT and KO mice were affected by MDMA administration only with aging. In middle-aged mice, MDMA administration induced a significant decrease in the density of TH-positive fibers in the dorsal striatum and SNc in both male and female WT and *Rhes* KO mice. Interestingly, the decrease observed in the dorsal striatum of adult and middle-aged male *Rhes* KO mice was higher than that observed in WT mice. Furthermore, *Rhes* KO adult males showed a more pronounced astrogliosis in the dorsal striatum and microgliosis in the dorsal striatum and SNc as compared with WT and female *Rhes* KO animals. Finally, while adult female *Rhes* KO mice did not show glial activation as compared to WT, susceptibility for dopamine neuron increased with ageing, suggesting for females a lower vulnerability to neurotoxicity as compared to males. These data give support to the influence of Rhes in regulating the survival of dopaminergic neurons, as shown by Pinna et al. [15]. Overall, Rhes is able to influence the survival of the nigrostriatal pathway, making *Rhes* KO mice a suitable model to unveil molecular mechanisms potentially involved in the vulnerability to the midbrain dopaminergic neuronal loss, under both physiological and pathological processes.

## 6. Rhes Influences Autophagy and Mitophagy Processes

As a consequence of a variety of both physiological and pathological stressors, including nutrient deprivation, aging, increase of reactive oxidative species (ROS), loss of proteostasis and/or genome instability, cells normally implement a primary protective mechanism based on a lysosomal degradation pathway, called autophagy, able to ensue nutrient and energy homeostasis, as well as a cytoplasmic quality control process, called autophagy [74,75]. Together with microautophagy and chaperone-mediated autophagy, macroautophagy (commonly referred to as autophagy) represents the best characterized mechanism of degrading and recycling potentially harmful cytosolic components that, when affected, might be a causative factor for several pathologies, including neurodegenerative disorders [76,77,78]. Dysfunctional autophagy machinery has been thoroughly investigated, either in patients suffering from PD or animal models of this human disease, as revealed by a significant disruption of autophagic flux in midbrain SNc neurons [79,80,81]. Of interest, among the most specialized forms of autophagy, mitophagy plays a central role for the selective removal of damaged mitochondria, thus constituting a biological sensor for the maintenance of mitochondrial biogenesis and calcium homeostasis [82,83]. In this framework, novel and growing evidence posit that Rhes may act as a remarkable modulator of both autophagy and mitophagy, making this small molecule of great interest for neurological disorders. Accordingly, in vitro studies showed that Rhes binds to Beclin-1 and activates autophagic flux, by competitively loosening Beclin-1/Bcl-2 interaction, in a mTOR-independent manner, since the effect was still present in the presence of rapamycin [35]. On the other hand, Sharma and colleagues elegantly showed for the first time that Rhes co-localizes with lysosomes, and interacts with globular mitochondria, in primary striatal neurons, as well as striatal cell lines [40]. Moreover, in the presence of 3-NP, Rhes improved damaged mitochondria clearance, through the binding with the mitophagy receptor, Nix [40], thus raising the notion that Rhes protein might be considered as a striatal mitophagy ligand [84], with a relevant impact upon striatal neuronal vulnerability (Figure 3).

## 7. Conclusions

Since its early identification by Sutcliff’s group [1], the Ras-related family member, Rhes, has been attracting many researchers who work on different topics, thanks to the pleiotropic actions of this highly striatal-enriched protein, which make it a suitable molecular adaptor, under both physiological and pathological conditions. In this line, taking a cue from what we have discussed in the present review, we can draw a sort of general picture about Rhes functions. First, Rhes is a membrane-tethered GTP-binding protein which negatively modulates the cAMP/PKA signaling pathway in a PTX-sensitive manner, most likely strengthening G_αi_ activity and inhibiting N-type (Ca_V_2.2) calcium channels [4,21,22]. Moreover, Rhes expression is developmentally modulated by thyroid hormone, showing increasing mRNA levels between the perinatal phases in rodents, and reaching the highest amount in adulthood [7,13], which entails its potential involvement in alterations of relevance to thyroid hormone-dependent neurological disorders, including cretinism. Second, based on the higher abundance of *Rhes* transcript in the striatal dopaminoceptive MSNs and ChIs, several studies clearly documented a pivotal role of this small molecule in the modulation of both dopamine D_1_R- and D_2_R-dependent transmission [9,10,16,23,24]. Taken together, these findings indicate that *Rhes*, through the inhibition of the striatal cAMP/PKA pathway, acts as a physiological brake for the dopamine neurotransmission, hence allowing to consider it a putative pharmacological target to counteract addictive disorders. Third, Rhes has the ability to bind to and activate mTORC1 that, among several trophic processes, worsens L-DOPA-induced dyskinesia symptoms, as demonstrated in PD animal models [41,42]. Interestingly, lack of Rhes, by reducing striatal mTORC1 activity, is able to attenuate LID severity, and downregulate the striatonigral neurons’ activity (Figure 1) [31,50]. Such results encourage further studies about Rhes function, that can be considered a promising pharmacological target aimed at alleviating such motor disturbances, causing negligible adverse effects, when compared to more selective mTORC1 inhibitors (rapamycin or other rapalogs) which, rather, strongly inhibit protein synthesis and, therefore, are basically considered toxic compounds. Fourth, *Rhes* is localized in the nigrostriatal pathway and modulates the survival of TH-positive neurons [15]. In keeping with this, and together with the ability of Rhes to modulate autophagy and mitophagy pathways (Figure 3) [23,40,85], we can pinpoint Rhes as a putative key survival mediator of striatal vulnerability, and so, be allowed to address deeper investigations on this issue.

## Figures and Tables

**Figure 1 ijms-22-05326-f001:**
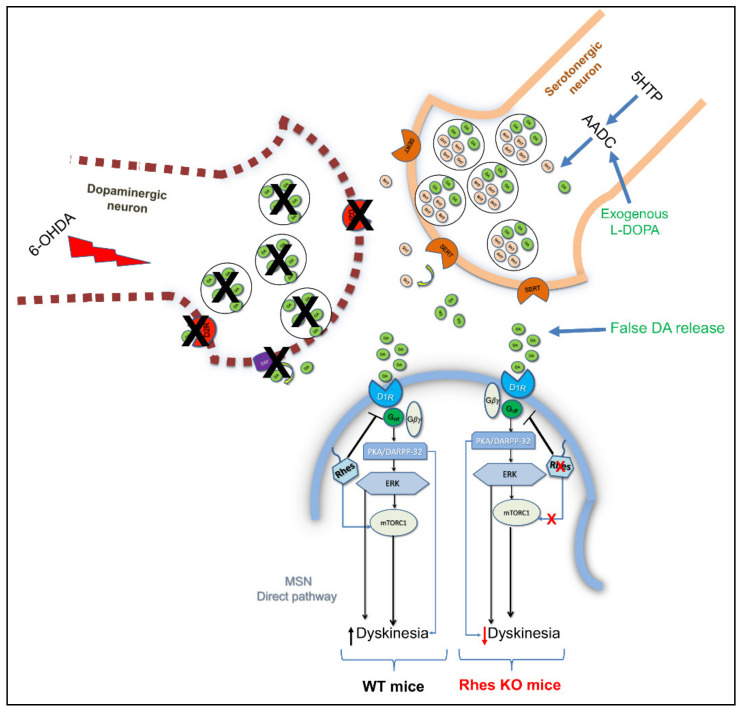
Rhes modulates L-DOPA-induced dyskinesia. Schematic representation showing that Rhes, following the activation of striatal mTORC1, mediates the dyskinetic effects triggered by L-DOPA administration in a 6-OHDA-lesioned mouse model, once converted to dopamine and released by serotonergic neurons in a non-physiological manner.

**Figure 2 ijms-22-05326-f002:**
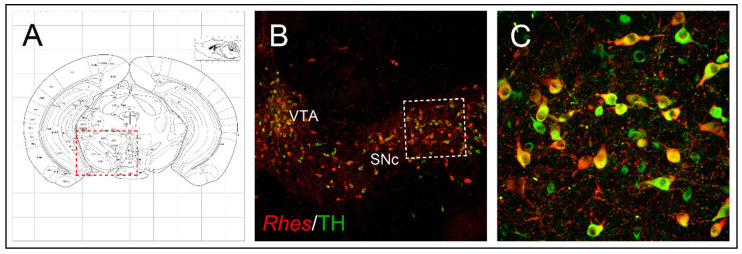
*Rhes* expression in midbrain dopaminergic neurons. (**A**) Schematic representation of a coronal section at the level of the midbrain. (**B**,**C**) Confocal images of brain coronal sections showing expression of *Rhes* in SNc and VTA TH-positive DA neurons. The figure has been adapted from Pinna et al. [15].

**Figure 3 ijms-22-05326-f003:**
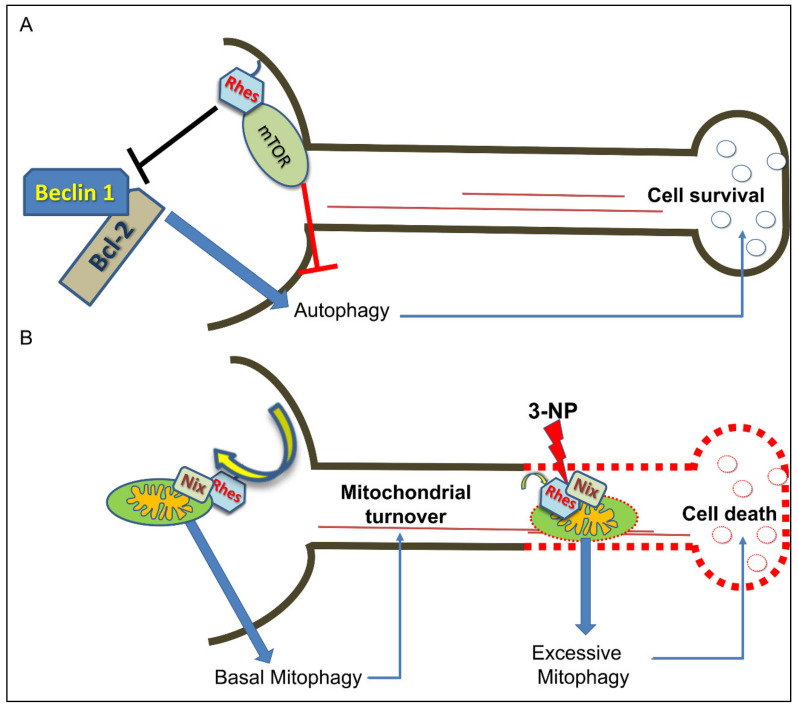
Rhes modulates striatal neuronal survival. (**A**,**B**) Working model, where Rhes is regarded as a key modulator of neuronal survival. (**A**) Rhes is able to bind to and activate mTOR, which normally inhibits autophagy. Moreover, it can also bind to Beclin-1 in particular cell conditions, hence displacing the inhibitory association between Bcl-2 and Beclin-1 that, eventually, activates autophagy in a mTOR-independent manner. Moreover, Rhes interacts with the mitophagy receptor, Nix, which drives autophagosomes to trigger basal mitochondrial degradation. (**B**) In the presence of mitochondrial toxin, 3-NP, such an interaction may bring about excessive mitophagy that, in turn, is able to promote neuronal cell death.

## Data Availability

Not applicable.

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
