# Peer review of "Involvement of the Protein Ras Homolog Enriched in the Striatum, Rhes, in Dopaminergic Neurons’ Degeneration: Link to Parkinson’s Disease"

_ijms, 2021, doi:10.3390/ijms22105326_

Round 1
Reviewer 1 Report
The review “Involvement of the protein Ras homolog enriched in the stria-2 tum, Rhes, in dopaminergic neurons degeneration: link to 3 Parkison’s disease” by Serra et al. provides a very deep revision of the protein Rhes and its signaling from its discovery to the present day.
The review is very complete, and the information and content is up to date. The expertise is clear in the treatment of the topics covered, and there is not much this reviewer could comment on the content. Nevertheless, I would like to provide some feedback in form of minor comments about some aspects of this review, for the consideration of the authors:
At the end of section 1, when describing the localization of Rhes RNA in human brains, it seems limited to cortex and limbic system, there is no explicit indication of the expression of Rhes in substantia nigra or VTA, and only a subtle reference to MSNs and ChIs neurons. Later in the article (lines 98ss) the authors reference observations of RNA expression in the striatum of PD patients. This contrast with the expression in rodent brains. As these nuclei are quite important in PD, would it be useful for the readers of the review to clarify this?
In section 7, it might result a bit confusing the results of the experiments done by Costa et al. (references 64 and 76): the authors indicate that “in a recent pioneering study” Costa et al found that lack of Rhes leads to a decrease in TH immunoreactivity in age, and they only found differences in glial cells comparing male and female. This is not so different from what people from the same lab had found previously, when “lack of Rhes led to a mild, although significant, reduction of midbrain TH-positive neurons in both 6- and 12-month-old KO mice [15].” But it gets a bit more confusing later, when they found that “In adult males, MDMA administration induced in both WT and KO animals a decrease of TH-positive fiber density in the dorsal striatum, as well as of the total number of TH-positive neurons in SNc.”, but “though a similar phenotype was observed in the SNc of WT females, TH-positive fibers and neurons seemed to be unaffected by MDMA in Rhes KO mice.” Does this mean that MDMA is protecting TH fibers and neurons in young adult female Rhes KO mice? I can understand that reading the original articles I could understand it, and trying to summarize these in the review will be a bit complicated.
Figure 3 legend seem to make references to figure 3C, but there is no figure 3C?
Also for Figure 3, could the picture be modified to illustrate the point of the figure more clearly? You can illustrate only once the relationship between Rhes, MTOR, Beclin and Bcl-2 (specially if it is not changed between top (A) and bottom (B) panel, it seems to me a bit distracting the presence of unnecessary elements (e.g. nuclei, dopamine receptors? Although not sure if there is a point to these), or confusing that 3-NP seems to be acting in Fig 3B in a previously affected mitochondria, although both mitochondria in Fig 3B are seen affected now, and only one in the Fig 3A above. What I mean, although the figure can be understood if you pay attention, it is not as straightforward to me as I am used to. So, would it be possible to simplify it but convey the same meaning?
General comments
All the sections are very complete and provide a lot of information, and this is great. But are written in only one paragraph. In my opinion it would be good for the reader to subdivide the paragraph in each section, to make it easier to read, if possible.
In line with this, the figures provided are very illustrative and appreciated. As a reader I would advocate for additional figures, especially for the information-dense description of pathways mentioned in each section, or reference some other source of information, if the authors consider that this would be helpful and is allowed for this review.
There are many instances where the authors indicate a relationship between Rhes activity (or other molecules) and its effect, (e.g. “Rhes interacts and modulates the PI3K/Akt/mTOR pathway”, “a direct influence of Rhes on RasGRP1-dependent signaling in affecting LID magnitude has been reported”, “the ability of Rhes to affect in vitro the drug-stimulated activation of the dopamine type 1 receptor”,…)“ but the effect is not clear from the language used: Does it increases, or decreases?, how much?,… Although not always possible, I would encourage the authors to indicate the effect whenever possible using verbs such as inhibit/activate, increase/reduce….
Very, very minor comments:
The sentence “Indeed, mice lacking Rhes gene showed an abnormal higher motor response to this psychostimulant in Rhes KO mice, than WT-treated animals” (lines 136-137) has some redundancies that might be edited.
In line 230 seem to be a typo, are the authors referencing Figure 1?
In lines 247-248, check grammar for “As a behavioral correlate to what morphologically observed”, seems to lack an auxiliary verb
In lines 309-311, “As a consequence of a variety of both physiological and pathological stressors, including nutrient deprivation, aging, increase of reactive oxidative species (ROS), loss of proteostasis, genome instability, cells normally implement a primary protective mechanism”, would you consider placing and/or between “proteostasis” and “genome”?
In lines 311-313, “cells normally implement a primary protective mechanism based on a lysosomal degradation pathway, able to ensue nutrient and energy homeostasis, as well as cytoplasmic quality control process, called autophagy”, would you consider placing “called autophagy” after “a lysosomal degradation pathway”? Or autophagy is only referred to the cytoplasmic quality control process?
Reviewer 2 Report
Serra and colleagues provide a well written and comprehensive review of Rhes role in the pathophysiology of Parkinson's disease. My only somewhat minor comment is that the authors provide more discussion about the potential clinical impact of Rhes for both prognostic evaluation and treatment of Parkinson's disease. While that might be premature, one question I have, for example, are there any studies that have attempted to modulate the Rhes expression in patients to corroborate the validity of the rodent based studies for the beneficial effect of Rhes in minimizing the side effects of prolonged L-DOPA treatment?
